# Proximal Composition and Nutritive Value of Raw, Smoked and Pickled Freshwater Fish

**DOI:** 10.3390/foods9121879

**Published:** 2020-12-17

**Authors:** Konrad Mielcarek, Anna Puścion-Jakubik, Krystyna J. Gromkowska-Kępka, Jolanta Soroczyńska, Sylwia K. Naliwajko, Renata Markiewicz-Żukowska, Justyna Moskwa, Patryk Nowakowski, Maria H. Borawska, Katarzyna Socha

**Affiliations:** Department of Bromatology, Faculty of Pharmacy with the Division of Laboratory Medicine, Medical University of Bialystok, Mickiewicza 2D Street, 15-222 Bialystok, Poland; anna.puscion-jakubik@umb.edu.pl (A.P.-J.); krystyna.gromkowska.kepka@gmail.com (K.J.G.-K.); jolanta.soroczynska@umb.edu.pl (J.S.); sylwia.naliwajko@umb.edu.pl (S.K.N.); renata.markiewicz@umb.edu.pl (R.M.-Ż.); justyna.moskwa@umb.edu.pl (J.M.); patryk.nowakowski@umb.edu.pl (P.N.); bromatos@umb.edu.pl (M.H.B.); katarzyna.socha@umb.edu.pl (K.S.)

**Keywords:** cluster analysis, energy value, freshwater fish, near infrared spectroscopy, proximal composition, Reference Intake

## Abstract

The aim of the study was to assess protein, fat, salt, collagen, moisture content and energy value of freshwater fish purchased in Polish fish farms. Eight species of freshwater fish (raw, smoked, pickled) were assessed by near infrared spectroscopy (NIRS). The protein content varied between 15.9 and 21.7 g/100 g, 12.8 and 26.2 g/100 g, 11.5 and 21.9 g/100 g in raw, smoked and pickled fish, respectively. Fat content ranged from 0.89 to 22.3 g/100 g, 0.72 to 43.1 g/100 g, 0.01 to 29.7 g/100 g in raw, smoked and pickled fish, respectively. Salt content ranged from 0.73 to 1.48 g/100 g, 0.77 to 3.39 g/100 g, 1.47 to 2.29 g/100 g in raw, smoked and pickled fish, respectively. A serving (150 g) of each fish product provided 53.2–71.9% of the Reference Intake (RI) for protein, 2.21–60.3% of the RI for fat, 21.3–61.3% of the RI for salt and 6.27–24.4% kJ/6.29–24.5% kcal of the RI for energy. Smoked fish had a higher protein and also fat content than raw and pickled fish, while smoked and pickled fish had higher salt content than raw fish. Cluster analysis was performed, which allowed to distinguish, on the basis of protein, fat, salt, collagen and moisture content, mainly European eel.

## 1. Introduction

Fish are one of the most valuable components of the human diet. According to the World Health Organization guidelines, it is recommended that fish is consumed two to three times a week in order to prevent lifestyle diseases [1]. Nevertheless, toxicologists recommended cautions because fish is an important source of exposure to many contaminants [2,3]. There are many essential nutrients in fish meat. Fish meat can be considered as a source of easily digestible protein, characterized by the content of essential amino acids for humans. In addition to protein, fish meat contains essential unsaturated fatty acids, especially omega-3 (eicosapentaenoic acid (EPA) and docosapentaenoic acid (DHA)), which improve human health and prevent many diseases. Omega-3 acids prevent heart disease and hypertension and also have an anti-atherosclerotic effect. They have been proven to reduce mortality of patients with coronary artery disease [4].

Warmia and Mazury (the Masurian Lake District) is a tourist destination which is highly popular with both Polish and foreign holiday makers. The area is famous for its 2000 lakes which are connected by rivers and canals. The region boasts a great number of producers of high quality, locally sourced food, who utilize traditional methods of food manufacture. Fish and fish products constitute the main branch of food production in the region [5]. In addition to raw and smoked freshwater fish, fish marinated in spirit vinegar (packed in glass jars) is also available in Warmia and Mazury. However, there is a lack of literature reports on the nutritional value of pickled freshwater fish. In terms of fishing typology, the division of lakes into five categories is commonly used in Poland: whitefish, bream, zander, rope-pike and crucian lake. One of the most frequently caught fish species in Masurian lakes and also commercially available, are: brown trout (*Salmo trutta morpha lacustris* L.), common bream (*Abramis brama* L.), common perch (*Perca fluviatilis* L.), common roach (*Rutilus rutilus* L.), common whitefish (*Coregonus lavaretus* L.), European eel (*Anguilla anguilla* L.), pike-perch (*Sander lucioperca* L.) and vendace (*Coregonus albula* L.) [6,7].

Near infrared spectroscopy (NIRS) is a rapid, sensitive and non-destructive instrumental method. NIRS does not require reagents and does not produce waste. It provides information on the molecular bonds of nutrient compounds and tissue structure in a scanned sample with minimal or no sample preparation [8]. In contrast to analysis of individual functional groups in mid-infrared spectroscopy, for the predictive capabilities of NIRS to be utilized, qualitative and quantitative calibration needs to be performed prior to the commencement of analysis (database creation) [9].

Over the last few years, NIRS has become a very common technique, frequently used to assess quality of food samples and adulteration including cocoa, coffee, milk or pork [8,10,11,12]. This method is also used to determine quality of fish and derived products: fatty acids in salmon oil, confirming authenticity of fish fillets and patties, histamine in tuna fish, sensory properties of Thai fish sauces, lipids in frozen fish, microbial content in salmon, distinction between wild and farmed sea bass [13,14,15,16,17,18,19].

The aim of the study was to assess and compare the proximal composition of freshwater fish products from the Masurian Lake District measured by NIRS and to estimate the nutritive value of one serving of raw, smoked and pickled fish (150 g) and also to consider using the NIRS method for studies of wild freshwater fish, in which meat can have very different nutrition composition, because of non-standardized and unregulated aquaculture.

## 2. Materials and Methods 

### 2.1. Research Areas and Sampling

The subject of the study were eight species of freshwater fish that live in the lakes of Warmia and Mazury Region, Poland (*n* = 212). The following fish species (raw (R), smoked (S), pickled (P)) were selected for the study: brown trout (*Salmo trutta morpha lacustris* L., S: *n* = 10; average weight: 7.5 kg), bream (*Abramis brama* L., R: *n* = 14, S: *n* = 10, P: *n* = 10; average weight: 1.5 kg), perch (*Perca fluviatilis* L., R: *n* = 10, S: *n* = 10, P: *n* = 10; average weight: 0.85 kg), roach (*Rutilus rutilus* L., R: *n* = 10, P: *n* = 12; average weight: 0.6 kg), whitefish (*Coregonus lavaretus* L., R: *n* = 10, S: *n* = 12, P: *n* = 10; average weight: 1.8 kg), eel (*Anguilla anguilla* L., R: *n* = 10, S: *n* = 13, P: *n* = 10; average weight: 1.8 kg), pike-perch (*Sander lucioperca* L., R: *n* = 10, P: *n* = 10; average weight: 8.5 kg) and vendace (*Coregonus albula* L., R: *n* = 10, S: *n* = 11, P: *n* = 10; average weight: 0.75 kg). Fish were collected from fishing farms in the towns and cities of the Masurian Lake District (Mikołajki, Mrągowo, Olsztyn, Orzysz, Pisz, Piękna Góra, Ruciane Nida). Fishing farms were located on the Czoś, Nidzkie, Niegocin, Roś, Śniardwy, Ukiel lakes. Raw, smoked (hot method) and marinated in spirit vinegar fish were purchased from fishing farms in the years: 2017 and 2018. When it comes to marinated fish products, it means fish that first have undergone heat treatment (baked, boiled, fried or smoked) and then added salt, sugar, food gelatin, vegetables and seasoning. After this pickled in a solution of spirit vinegar. However, not all studied fish were commercially achievable in all forms (R, S, P). Brown trout (R, P), common roach (S) and pike-perch (S) were unavailable to purchase.

### 2.2. Fish Samples Preparation Procedure

The Association of Official Analytical Chemists (AOAC) Official Method 983.18 (Meat and Meat Products Preparation of Sample Procedure), was used for samples preparation [20]. Firstly, fish meat were separated from other parts, which are inedible. The preparation of marinated fish was varied. The jelly was removed from the meat. After this step, a mechanical homogenizer was used to homogenize samples (IKA T 18 digital, ULTRA-TURRAX, IKA^®^-Werke GmbH and CO. KG, Staufen, Germany). An approximately 180 g ground sample was placed in a 140 mm round sample dish and the dish was placed in the FoodScan spectrophotometer (FoodScan, FOSS, Hilleroed, Denmark).

### 2.3. Determination of Protein, Fat, Salt, Collagen and Moisture Content in Fish Samples and Energy Value by Near Infrared Spectroscopy Method (NIRS)

The content of each component in fish samples was determined using the NIRS method, based on a calibration model created using an Artificial Neural Network (ANN) and a database for determining protein, fat, salt, collagen and moisture content in meat products. The calibrations was created by correlating the numerical results for each parameter in the sample with the near infrared spectrum. Depending on the type of product, calibration covers from several dozen to several hundred samples. Prior to determining the content of the tested component, accuracy and repeatability were checked by performing a calibration test. Calibrations and ANNs developed by FOSS were used for the study. A homogenized sample was brought to 20 ± 0.5 °C and the scanning process was initiated. The measurement covered the 0.85–1.05 µm range. Each measurement was performed three times. In between laboratory studies it was shown that RSDr (relative standard deviation) for protein ranged from 0.54 to 5.23, for fat: from 0.52 to 6.89%, for moisture: from 0.39 to 1.55%, which confirms the reproducibility of the method [21]. Results were rounded to 2 decimal places and displayed as g/100 g of protein, fat, salt, collagen and moisture. Energy value was calculated assuming the following conversion factors: 4 kcal/17 kJ and 9 kcal/37 kJ for 1 g of proteins and 1 g of lipids, respectively [22].

### 2.4. Statistical Analysis

The Statistica v.13.3 software (TIBCO Software Inc., Palo Alto, CA, USA) was used to perform the statistical analysis. Assessment of normality was conducted using the Kolmogorov-Smirnov test. Differences between independent groups were tested using the Student *t*-test. Results were considered statistically significant when *p* < 0.05. Percentages of the RIs for energy, protein and fat for an adult were calculated in relation to the consumption of 150 g of fish meat, which is the assumed average fish fillet weight [23]. In order to indicate the similarities in the composition between the studied species of fish, a Cluster Analysis (CA) was performed. The single bond principle was used as the method of agglomeration and the Euclidean distance was used as a measure of distance.

## 3. Results and Discussion

### 3.1. Proximal Composition of Fish Product Samples

The majority of studies that utilize the NIRS method involve scanning food samples followed by chemometric analysis. Software used in our research provided information on the composition of a sample on the basis of the created database and artificial neural networks. In the present study, the NIRS method was used for the first time to assess the quality of edible parts of freshwater fish from Poland. The results regarding content of the studied parameters in edible parts of fish species are presented in (Figure 1, Figure 2, Figure 3, Figure 4 and Figure 5). In the study, in terms of the content of each component between different raw, smoked and pickled fish species, were found a statistically significant differences.

Protein content in the studied fish form products is presented in (Figure 1 and Table 1). The total average protein content in raw fish was 19.61 ± 1.2 g/100 g. Among all studied raw fish, the highest average amount of protein was found in the whitefish (20.75 ± 0.9 g/100 g), while the lowest protein content was revealed in eel (17.72 ± 1.0 g/100 g). Kiczorowska et al. [24] investigated protein content of raw bream and whitefish from fish farms in the south and east of Poland. The amount of protein was 19.8 ± 0.2 g/100 g and 19.4 ± 0.3 g/100 g, respectively. Jankowska et al. [25] examined the composition of perch from Lake Dgał Wielki (Masurian Lake District, northern Poland) and found a marginally lower average protein content (17.66 ± 0.2 g/100 g) than in our study (19.33 ± 0.6 g/100 g). Łuczyńska et al. [26] studied the composition of perch from Mosąg and Wadąg lakes in central Poland. The average protein content was 17.66 ± 1.56 g/100 g and 18.49 ± 2.14 g/100 g, respectively, which is also lower than that reported in the present study. Polak-Juszczak and Adamczyk [27] investigated freshwater fish from the Vistula Lagoon and reported protein contents similar to those found in the present study: bream (18.1 ± 0.5 g/100 g), eel 14.5 ± 0.8 g/100 g, perch (18.1 ± 0.5 g/100 g), pike-perch (19.5 ± 0.5 g/100 g), roach (18.3 ± 0.4 g/100 g). Khalili et al. [28] also studied freshwater fish (lakes located in Czech Republic). All tested fish showed a level of protein like our study but it was lower than in Polish fish (17.1 ± 1.55 to 19.2 ± 2.20 g/100 g): bream (18.0 ± 1.24 g/100 g), perch (17.6 ± 1.85 g/100 g) and brown trout (19.2 ± 1.50 g/100 g). Having compared the results of other authors to those obtained in the present study, we noticed that a higher protein content of 21.2 ± 0.1 g/100 g was demonstrated by Lee et al. [29] in freshwater rainbow trout from fish farms located in South Korea. Fallah et al. [30] studied the composition of freshwater fish caught in a river in Iran. The average protein content was 20.9 ± 0.40 g/100 g. A higher average protein content was also shown by Badiani et al. [31] in freshwater sturgeon samples obtained from an intensive commercial fish farm in northern Italy (19.23 ± 0.17 g/100 g). However, Bayse et al. [32] demonstrated that in American shad, an anadromous fish from Connecticut River (USA), the average protein content was marginally lower (16.4 ± 4.7 g/100 g). In addition, Chandrashekar and Deosthale [33] showed that the average protein content in Indian freshwater fish was 15.8 ± 1.93 g/100 g. An even lower concentration was recorded by Bogard et al. [34] in freshwater fish in Bangladesh (from 11.9 to 20.5 g/100 g).

The total mean protein content in smoked fish samples was 21.96 ± 2.7 g/100 g. Out of all tested samples of smoked fish, vendace (23.97 ± 1.5 g/100 g) had the highest mean protein content and eel the lowest (18.32 ± 3.0 g/100 g). The total average amount of protein in marinated fish was 17.22 ± 2.7 g/100 g. The highest concentration was found in common whitefish (19.56 ± 1.9 g/100 g) and the lowest in eel (13.12 ± 1.2 g/100 g). Kiczorowska et al. [24] investigated protein in smoked whitefish and bream (Poland). The content was 22.3 ± 0.2 g/100 g and 22.6 ± 0.4 g/100 g, respectively. There are no literature data regarding protein content in pickled freshwater fish from Poland.

Differences in protein content may result, among others, from different fishing locations and timings. Efficiency in using dietary protein depends on its quantity and nutritional value [35]. Protein quality is primarily determined by digestibility, which is highest in the case of fish (98.3–98.8%), lower for meat (97.5%) and still lower for milk (95.5%) [27,36]. High quality of fish protein is also due to the fact that it is rich in exogenous amino acids. Polak-Juszczak and Adamczyk [27] showed that fish from the Vistula Lagoon had a higher content of all essential amino acids than the reference protein.

Fat concentration in fish is presented in (Figure 2 and Table 2). The total average amount of fat in raw fish was 4.76 ± 5.6 g/100 g. The highest average amount of fat in raw fish was found in eel (18.2 ± 2.5 g/100 g). The lowest average amount of fat was in pike-perch meat (1.11 ± 0.1 g/100 g). Jankowska et al. [25] studied proximate composition of perch and reported fat content of 0.3 ± 0.0 g/100 g, which was almost 10 times lower than in the present study. Łuczyńska et al. [26] investigated proximate composition of perch from lakes in central Poland. The mean fat content was 0.50 ± 0.38 g/100 g. Polak-Juszczak and Komar-Szymczak [37] studied the total fat composition of freshwater fish from the Vistula Lagoon. The mean fat content was 3.14 ± 0.78 g/100 g in bream, 0.79 ± 0.2 g/100 g in roach, 0.46 ± 0.04 g/100 g in perch. Polak-Juszczak and Adamczyk [27] investigated freshwater fish from the Vistula Lagoon and reported fat contents similar to those found in the present study: bream (2.59 ± 1.26 g/100 g), eel (28.9 ± 4.90 g/100 g), perch (0.12 ± 0.05 g/100 g), pike-perch (0.13 ± 0.11 g/100 g), roach (0.56 ± 0.34 g/100 g). Lee et al. [29] revealed that fat content of freshwater rainbow trout from fish farms in South Korea was 3.6 ± 0.2 g/100 g. Aggelousis and Lazos [38] demonstrated that fat content of freshwater fish from Greece ranged from 0.6 ± 0.21 to 3.5 ± 0.65 g/100 g and fat content of bream was 1.0 ± 0.48 g/100 g. Suloma et al. [39] studied freshwater fish from Philippines and reported fat content from 2.54 ± 0.27 to 19.51 ± 1.78 g/100 g in each species. Badiani et al. [28] reported mean fat content to be 7.63 ± 0.58 g/100 g. Bayse et al. [32] showed that average fat content of freshwater fish was 6.2 ± 3.1 g/100 g. Bogard et al. [34] revealed that fat content ranged from 0.30 to 18.3 g/100 g. Chandrashekar and Deosthale [33] demonstrated mean fat content of freshwater fish to be 1.0 ± 0.35 g/100 g. Fallah et al. [30] showed that mean fat content was 5.01 ± 0.35 g/100 g in freshwater fish. Khalili et al. [28] investigated freshwater fish from the Czech Republic and revealed that fat content ranged from 0.74 ± 0.04 g/100 g in perch to 4.04 ± 0.81 g/100 g in common nase.

Among all tested smoked fish, the eel had the highest mean fat concentration (28.16 ± 8.6 g/100 g) and perch the lowest (1.03 ± 0.2 g/100 g), while the mean fat content was 10.54 ± 11.2 g/100 g. Branciari et al. [40] investigated smoked tench (*Tinca tinca*) pâtés from Trasimeno Lake (Italy). Fat content of this freshwater species was 4.69 g/100 g. Glew et al. [41] studied fish (smoked, freshwater) from Northern Nigeria. Fish were divided into head, midsection and tail sections. Fatty acid content of each of the three sections of the studied fish was very similar. Overall, fatty acid levels ranged from 9.43 to 11.5 g/100 g.

The total average amount of fat in marinated fish was 6.11 ± 8.0 g/100 g, with the highest concentration of 25.04 ± 4.1 g/100 g revealed in eel and the lowest concentration of 1.72 ± 2.0 g/100 g found in pike-perch samples. There is a lack of literature data regarding fat content in pickled freshwater fish from Warmia and Mazury Region (Poland) and also from other parts of world.

Salt concentration in fish is shown in (Figure 3 and Table 3). The total average amount of salt in raw fish was 1.03 ± 0.2 g/100 g. The highest average amount of salt from raw fish had the eel with 1.34 ± 0.1 g/100 g, while the lowest mean concentration (0.85 ± 0.1 g/100 g) was detected in common whitefish. The total average amount of salt in smoked fish was 1.76 ± 0.6 g/100 g. The highest average amount of salt from smoked fish had in perch (2.45 ± 0.5 g/100 g) and the lowest in brown trout (1.09 ± 0.2 g/100 g). In marinated fish, the total average amount of salt was 1.88 ± 0.2 g/100 g. The highest average salt content was discovered in perch (2.04 ± 0.2 g/100 g) and the lowest in eel samples (1.69 ± 0.1 g/100 g). Puścion-Jakubik et al. [42] studied salt content in smoked freshwater fish from the lakes of Warmia and Mazury Region in Poland and established that bream contained 1.87 ± 0.3 g/100 g salt, brown trout 2.09 ± 0.2 g/100 g, common whitefish 2.53 ± 0.3 g/100 g, eel 2.41 ± 0.2 g/100 g and vendace 2.71 ± 0.7 g/100 g. Salt contents from that study were almost like in our study. Branciari et al. [40] demonstrated that salt content in smoked tench pâtés was 0.56 g/100 g. There is limited literature data regarding salt content in freshwater fish.

The importance of checking salt content in foodstuffs must be emphasized here as sodium consumption needs to be closely monitored by individuals with particular health conditions, such as hypertension. In the case of these consumers, products with a reduced salt content, such as raw fish should be selected and prepared for consumption. The present study demonstrated that smoked and pickled fish had a higher salt content (added during the production process). The smoking of fish is done to improve sensory values: smell, taste and color. The quality of the final product is influenced by many factors including smoke composition and salt content. When choosing smoked fish, we should select hot smoked fish due to a lower salt content (from 1 to 3%) as cold smoked fish has a higher salt content (from 6 to 8%) [43].

Content of collagen in fish products are shown in (Figure 4 and Table 4). The highest average collagen concentration in raw fish was in the eel (0.75 ± 0.2 g/100 g) and also in meat of perch (0.75 ± 0.5 g/100 g), while the lowest was demonstrated in vendace (0.36 ± 0.4 g/100 g). The total average collagen content in raw fish was 0.57 ± 0.3 g/100 g. Among all studied smoked fish, eel had the highest mean collagen concentration (1.39 ± 0.6g/100 g) and common whitefish displayed the lowest (0.07 ± 0.1 g/100 g), while the mean collagen content was 0.54 ± 0.6 g/100 g. The total average collagen content in marinated fish was 1.30 ± 0.8 g/100 g, with the highest content of 2.45 ± 0.4 g/100 g found in eel and the lowest concentration of 0.36 ± 0.4 g/100 g found in common whitefish samples. There is no data in the literature, where are reports regarding collagen content in freshwater fish. A higher collagen content in pickled freshwater fish in comparison to raw and smoked fish is caused by adding food gelatin during the production process. 

Water content in the investigated freshwater fish products is presented in (Figure 5 and Table 5). Among all raw fish, the highest amount of moisture was found in the pike-perch with 78.4 ± 0.3 g/100 g, while the lowest moisture content was revealed in eel (64.4 ± 2.1 g/100 g). The total average moisture content in raw fish was 75.3 ± 4.7 g/100 g. The total average moisture content in smoked fish was 65.6 ± 10.0 g/100 g. Out of all samples of smoked fish, perch had the highest mean moisture content (73.6 ± 1.6 g/100 g), while eel had the lowest (50.5 ± 7.3 g/100 g). The total average moisture content in marinated fish was 70.3 ± 9.6 g/100 g, with the highest concentration of 74.8 ± 1.1 g/100 g found in common whitefish and the lowest concentration of 59.4 ± 3.1 g/100 g demonstrated in eel. Jankowska et al. [25] studied the composition of perch from Dgał Wielki Lake and reported moisture content of 80.9 ± 0.1 g/100 g. Łuczyńska et al. [26] demonstrated that moisture content in perch was 79.94 ± 1.1 g/100 g and 81.17 ± 0.87 g/100 (Lakes Mosąg and Wadąg in central Poland). Lee et al. [29] revealed that moisture content in freshwater rainbow trout was 75.3 ± 0.1 g/100 g. Aggelousis and Lazos [38] reported that in freshwater fish from Greece, variations in moisture content were small and most of the values were between 78 ± 0.6 and 80 ± 1.3 g/100 g. Badiani et al. [31] investigated freshwater sturgeon (Northern Italy) and found that the mean moisture content was 72.49 ± 0.54 g/100 g. Bayse et al. [32] demonstrated that moisture content in American shad (Connecticut River, USA) was 72.3 ± 4.7 g/100 g. Bogard et al. [34] studied proximate composition of freshwater fish in Bangladesh. Moisture content ranged from 60.2 to 85.4 g/100 g. Chandrashekar and Deosthale [33] investigated proximate composition of Indian freshwater fish. The mean moisture content was 80.1 ± 2.48 g/100 g. Fallah et al. [30] studied proximate composition of fish captured in a river in Iran. The mean moisture content was 70.4 ± 0.73 g/100 g. Branciari et al. [40] investigated smoked tench (*Tinca tinca*) pâtés (Trasimeno Lake, Italy). Moisture content of this freshwater fish species was 69.96 g/100 g.

### 3.2. Correlations Between Parameters Determined by the NIRS Method

Correlations between individual parameters for raw, smoked and pickled fish were tested. A practically full negative correlation was found between fat and moisture content for fresh (*r* = −0.98) and smoked (*r* = −0.99) fish. A very high negative correlation was observed in smoked fish between protein and fat content (*r* = −0.81), protein and collagen content (*r* = −0.74) and between collagen and moisture content (*r* = −0.81). For pickled fish, a negative correlation was discovered only between collagen and protein content (*r* = −0.88). A very high positive correlation was found in smoked fish between protein and collagen content (*r* = 0.74) and between fat and collagen content (*r* = 0.84). Breck [44] showed a very strong correlation between moisture weight and protein weight in fish, which is related to physiological or biochemical aspects. The amount of moisture per unit of protein is smaller for larger fish. Research conducted as part of the present study demonstrated a practically full negative correlation between moisture and fat content.

### 3.3. Energy Value of Fish Product Samples

Energy values of the studied fish are shown in (Table 6). The mean energy value was 507.8 kJ/100 g and 121.3 kcal/100 g. Out of raw freshwater fish samples, eel had the highest energy value of 982.6 kJ/100 g and 234.7 kcal/100 g, while pike-perch the lowest (372.4 kJ/100 g and 89.0 kcal/100 g). Out of smoked freshwater fish samples, the highest energy value was found in eel with 1367.9 kJ/100 g and 326.7 kcal/100 g, while the lowest in perch (420.5 kJ/100 g and 100.4 kcal/100 g). The total energy value of smoked freshwater fish samples was 764.9 kJ/100 g and 182.7 kcal/100 g. The highest energy value of pickled freshwater fish samples was found in eel with 1163.3 kJ/100 g and 277.8 kcal/100 g, while the lowest in perch (351.3 kJ/100 g and 83.9 kcal/100 g). The total energy value of pickled freshwater fish samples was 518.6 kJ/100 g and 123.9 kcal/100 g. Kiczorowska et al. [24] studied energy value of raw and smoked bream and common whitefish. Raw fish had 371 ± 0.41 kJ/100 g, 88.5 ± 0.32 kcal/100 g and 371 ± 0.13 kJ/100 g, 88.6 ± 0.25 kcal/100 g, respectively. Smoked fish had 416 ± 0.27 kJ/100 g, 99.3 ± 0.26 kcal/100 g and 418 ± 0.17 kJ/100 g, 99.9 ± 0.39 kcal/100 g, respectively. The results were lower than those obtained in our study.

### 3.4. Calculated Reference Intake for Energy Value and Nutrients Based on the Consumption of Raw, Smoked and Pickled Fish (One Serving—150 g)

Percentages of the RIs for kcal, kJ, protein, fat and salt met by consuming one serving (150 g) of fish products groups are presented in (Table 7). A 150 g portion of raw fish meat gives from 6.65 to 17.5%, smoked fish from 7.51% to 24.4% and pickled fish from 6.27 to 20.8% of RI (kJ). For kcal, a portion of raw fish gives from 6.67 to 17.6%, smoked fish from 7.53 to 24.5% and pickled fish from 6.29 to 20.9% of the RI values. Taking into account protein content, 150 g of raw fish provides from 53.2 to 62.3%, smoked from 55.0 to 71.9% and marinated from 39.4 to 58.7% of the RI for protein. Our study shown that the consumption of raw fish provides from 2.38 to 39.0% of the RI for fat (150 g). 150 g of meat of smoked fish gives from 2.21 to 60.3% of the RI for fat, while marinated fish gives from 3.69 to 51.5% of the RI for fat. In terms of salt, a 150 g portion of raw fish provides from 21.3 to 33.5% of the RI. The same amount of meat of smoked fish covers from 27.3 to 61.3% of the RI for salt. 150 g of marinated fish covers from 42.3% to 51.0% of the RI for salt [23].

Kiczorowska et al. [24] established that consuming one serving of raw or smoked bream covers 8.85% kcal/ 9.93% kJ of the daily energy requirement and common whitefish covers 8.86% kcal/ 9.99% kJ. Similar values were obtained in the present study. Similarly to our results, Bogard et al. [34] demonstrated that energy content of freshwater fish from Bangladesh ranged from 267 to 1020 kJ/100 g. Khalili et al. [28] who investigated energy value of fish from the Czech Republic lakes, had a similar results. In their study, bream had 528 ± 18 kJ/100 g and 126 ± 4 kcal/100 g, perch had 500 ± 31 kJ/100 g and 114 ± 2 kcal/100 g, brown trout had 619 ± 56 kJ/100 g, 148 ± 13 kcal/100 g. Ramos et al. [45] investigated energy value of freshwater fish from the Brazilian Pantanal. Energy value ranged from 294.3 to 692.8 kJ/100 g. Branciari et al. [40] investigated smoked tench (Tinca tinca) pâtés of freshwater species and established that energy value of this product was 133 kcal/100 g and 556 kJ/100 g. Analysis of the nutritional value of fish is important since, according to the Central Statistical Office [46] data, average fish consumption in Poland has increased, that is, in 2016 it was 12.9 kg/person/year, which is 3.1% higher than in the previous year. However, sea fish is more frequently consumed in Poland (70% of consumers) in comparison to freshwater fish (30%) [47,48].

### 3.5. Cluster Analysis (CA)

The results of the CA are presented in the form of a dendrogram (Figure 6) shows data for raw, (Figure 7) for smoked and (Figure 8) for pickled fish. Dendrograms show similarities in protein, fat, salt, collagen and water content in the types of fish tested. The criteria used made it possible to distinguish a cluster including European eel (raw, pickled), while in the terms of smoked fish—the cluster included European eel and Brown trout.

## 4. Conclusions

Our study revealed that smoked fish had a higher protein content than raw and pickled fish, while smoked and pickled fish had a higher salt content than raw fish, which underlines the fact that heat treatment significantly affects the nutritional value of fish. In the case of bream, eel and vendace, a higher average fat content in smoked fish was noted. The increase can be explained by water loss during the smoking process. A lower moisture content in smoked fish, compared to raw fish, was reported for all tested species. Cluster analysis could be useful in differentiation of the types of fish (especially European eel) based on their primary composition—protein, fat, salt, collagen and moisture content. Moreover, the NIRS method creates a number of possibilities that allow for a quick and accurate examination of the quality of fish and fish products. The advantage of using NIR analysis is that it ensures rapid analysis data for better decision making in food and agri-production processes. Compared to traditional analysis methods it requires little or no sample preparation (only mechanical homogenization) and no chemicals or consumables. It is non-destructive, operator friendly, fast (about 50 seconds per one sample), reliable and precise. NIR food analysis method helps choose the most optimal way of fish processing in terms of nutritional preservation (smoking or pickling). This method is also, helpful for analyzing all stages of fish production—from checking incoming raw material to final product control.

An ANN is a calibration model, which NIR method use. A major advantage with ANN compared to other calibration methods is that it covers a big range of parameters, without need to switch between many individual calibrations. For instance, when testing fish meat, the analytics can stay with the same calibration for different fat ranges and different products instead of having to decide on changing to another calibration and then implementing it on the instrument. Where ever they are used, ANN calibrations save time and make life easier for the operator while also reducing the risk of operator error. Use ANN for analysis, also influences to fewer calibrations to verify, minimizing the validation time and costs involved. When many products and parameters are involved, this aspect can save significant costs compared to using other calibration methods. For this reason, this method is very useful for studies of wild freshwater fish, in which meat can have very different nutrition composition, because of non-standardized and unregulated aquaculture.

## Figures and Tables

**Figure 1 foods-09-01879-f001:**
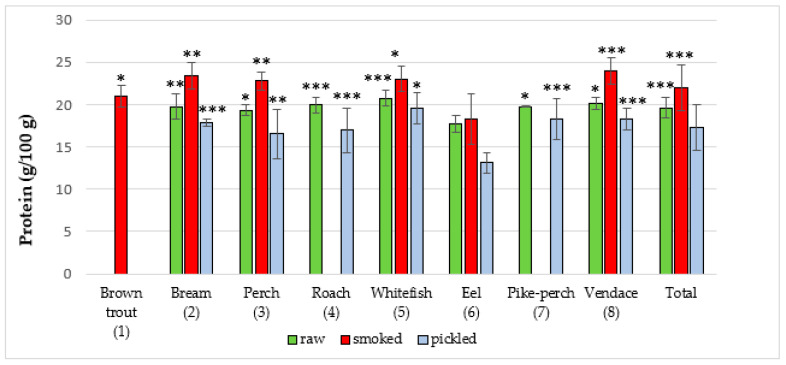
Protein content in raw (**R**), smoked (**S**) and pickled (**P**) fish products samples (g/100 g). The statistical analysis has shown significant differences of content of protein between species in R, S and P groups of freshwater fish products (*** *p* < 0.001, ** *p* < 0.01, * *p* < 0.05). R: *p* < 0.05 (8 vs. 3, 6) and (7 vs. 3, 6); *p* < 0.01, (2 vs. 6); *p* < 0.001 (3 vs. 6), (4 vs. 6) and (5 vs. 3, 6, 7); S: *p* < 0.05, (1 vs. 6); *p* < 0.01, (2 vs. 1, 6), (3 vs. 1, 6) and (5 vs. 1, 6); *p* < 0.001, (8 vs. 1, 6); P: *p* < 0.05, (5 vs. 2, 3, 4, 6); *p* < 0.01, (3 vs. 6); *p* < 0.001, (2 vs. 6), (4 vs. 6), (7 vs. 6) and (8 vs. 6); Total: *p* < 0.001, (R vs. P) and (S vs. P and R).

**Figure 2 foods-09-01879-f002:**
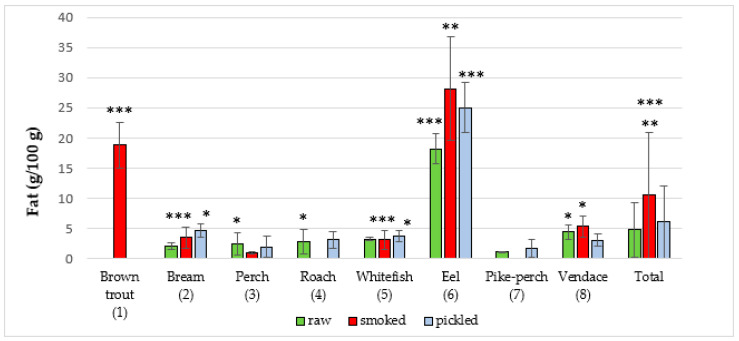
Fat content in raw (**R**), smoked (**S**) and pickled (**P**) fish products samples (g/100 g). The statistical analysis has shown significant differences of content of fat between species in R, S and P groups of freshwater fish products (*** *p* < 0.001, ***p* < 0.01, * *p* < 0.05). R: *p* < 0.05, (3 vs. 7), (4 vs. 7), (8 vs. 1, 2, 3, 4, 5, 7); *p* < 0.001, (6 vs. 1, 2, 3, 4, 5, 7, 8); S: *p* < 0.05, (8 vs. 2, 3, 5); *p* < 0.01, (6 vs. 1, 2, 3, 5, 8); *p* < 0.001, (1 vs. 2, 3, 5, 8), (2 vs. 3) and (5 vs. 3); P: *p* < 0.05, (2 vs. 3, 4, 5, 7, 8) and (5 vs. 3, 7); *p* < 0.001, (6 vs. 2, 3, 4, 5, 7, 8); Total: *p* < 0.01, (S vs. P); *p* < 0.001, (S vs. R).

**Figure 3 foods-09-01879-f003:**
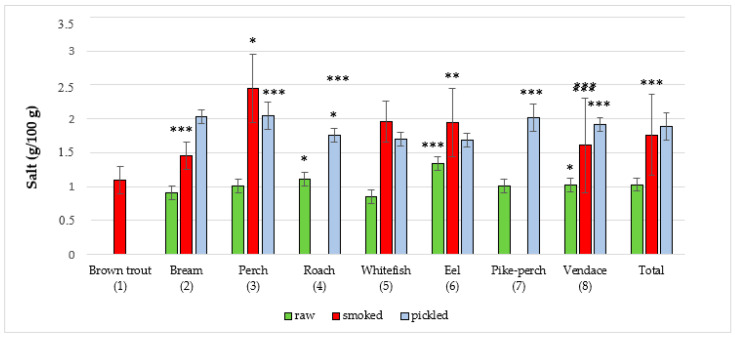
Salt content in raw (**R**), smoked (**S**) and pickled (**P**) fish products samples (g/100 g). The statistical analysis has shown significant differences of content of salt between species in R, S and P groups of freshwater fish products (*** *p* < 0.001, ** *p* < 0.01, * *p* < 0.05). R: *p* < 0.05, (3 vs. 2, 5), (4 vs. 2, 5, 7) and (8 vs. 2, 5); *p* < 0.001, (6 vs. 2, 3, 4, 5, 7, 8); S: *p* < 0.01, (3 vs. 1, 2, 5, 6, 7, 8); *p* < 0.01, (6 vs. 1, 2), *p* < 0.001, (2 vs. 1), (5 vs. 1, 2) and (8 vs. 1); P: *p* < 0.05, (2 vs. 4, 5, 6, 8); *p* < 0.01, (4 vs. 6); *p* < 0.001, (3 vs. 4, 5, 6), (7 vs. 4, 5, 6) and (8 vs. 4, 5, 6); Total: *p* < 0.001, (P vs. R) and (S vs. R).

**Figure 4 foods-09-01879-f004:**
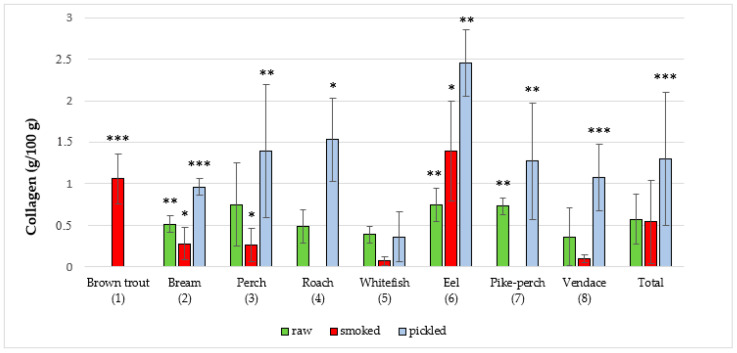
Collagen content in raw (**R**), smoked (**S**) and pickled (**P**) fish products samples (g/100 g). The statistical analysis has shown significant differences of content of collagen between species in R, S and P groups of freshwater fish products (*** *p* < 0.001, ** *p* < 0.01, * *p* < 0.05). R: *p* < 0.05, (3 vs. 5); *p* < 0.01, (2 vs. 5), (6 vs. 2, 4, 5, 8) and (7 vs. 2, 4, 5, 8); S: *p* < 0.05, (2 vs. 5, 8), (3 vs. 5, 8) and (6 vs. 2, 3, 5, 8); *p* < 0.001, (1 vs. 2, 3, 5, 8); P: *p* < 0.05, (4 vs. 2, 5, 8); *p* < 0.01, (3 vs. 5), (6 vs. 2, 3, 4, 5, 7, 8) and (7 vs. 5); *p* < 0.001, (2 vs. 5) and (8 vs. 5); Total: *p* < 0.001, (P vs. R and S).

**Figure 5 foods-09-01879-f005:**
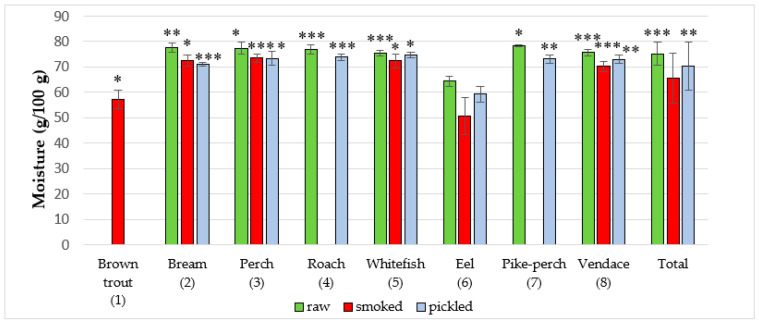
Moisture content in raw (**R**), smoked (**S**) and pickled (**P**) fish products samples (g/100 g). The statistical analysis has shown significant differences of content of moisture between species in R, S and P groups of freshwater fish products (*** *p* < 0.001, ** *p* < 0.01, * *p* < 0.05). R: *p* < 0.05, (3 vs. 5, 6) and (7 vs. 4, 5, 6, 8); *p* < 0.01, (2 vs. 5, 6, 8); *p* < 0.001, (4 vs. 6), (5 vs. 6) and (8 vs. 6); S: *p* < 0.05, (1 vs. 6), (2 vs. 1, 6, 8) and (5 vs. 1, 6, 8); *p* < 0.001, (3 vs. 1, 6, 8) and (8 vs. 1, 6); P: *p* < 0.05, (3 vs. 2, 6) and (5 vs. 2, 6, 7, 8); *p* < 0.01, (7 vs. 2, 6) and (8 vs. 2, 6); *p* < 0.001, (2 vs. 6) and (4 vs. 2, 6); Total: *p* < 0.01, (P vs. S), *p* < 0.001, (R vs. P and S).

**Figure 6 foods-09-01879-f006:**
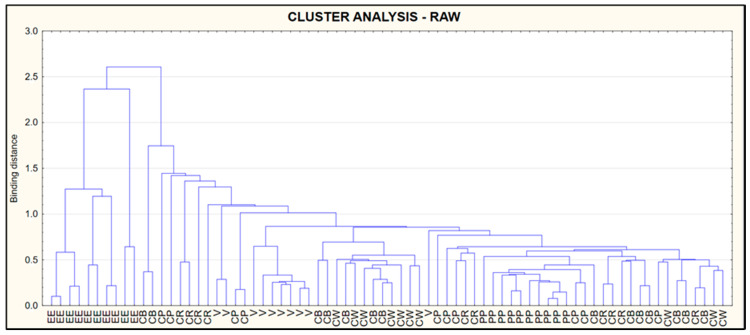
Dendrogram of similarity in protein, fat, salt, collagen and water in raw fish.

**Figure 7 foods-09-01879-f007:**
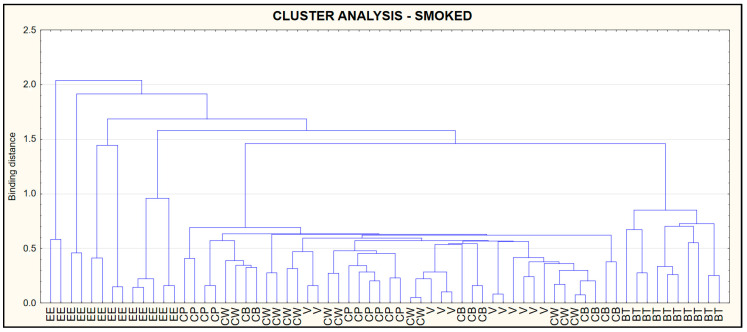
Dendrogram of similarity in protein, fat, salt, collagen and water in smoked fish.

**Figure 8 foods-09-01879-f008:**
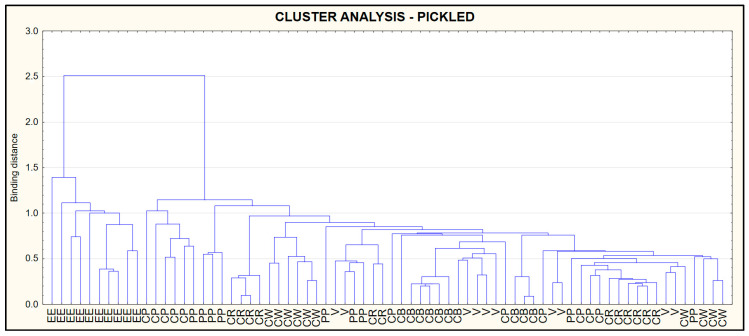
Dendrogram of similarity in protein, fat, salt, collagen and water in pickled fish.

**Table 1 foods-09-01879-t001:** Protein content in raw, smoked and pickled fish product samples (g/100 g).

Fish Species	Raw				Smoked				Pickled			
	n	Mean ± SD	Min.	Max.	n	Mean ± SD	Min.	Max.	n	Mean ± SD	Min.	Max.
Brown trout (1) (*Salmo trutta m.* *lacustris* L.)	NA ^a^	NA	NA	NA	10	20.98 ± 1.3	19.57	23.14	NA	NA	NA	NA
Common bream (2) (*Abramis brama* L.)	14	19.72 ± 1.5	16.35	21.12	10	23.40 ± 1.5	21.62	25.39	10	17.84 ± 0.4	17.29	18.53
Common perch (3) (*Perca fluviatilis* L.)	10	19.33 ± 0.6	18.47	19.90	10	22.79 ± 1.1	21.66	24.18	10	16.52 ± 2.9	12.22	20.25
Common roach (4) (*Rutilus rutilus* L.)	10	19.92 ± 0.9	18.8	21.13	NA	NA	NA	NA	12	16.96 ± 2.6	13.68	19.72
Common whitefish (5) (*Coregonus lavaretus* L.)	10	20.75 ± 0.9	19.3	21.74	12	23.01 ± 1.5	23.01	20.39	10	19.56 ± 1.9	16.62	21.85
European eel (6) (*Anguilla anguilla* L.)	10	17.72 ± 1.0	15.88	18.98	13	18.32 ± 3.0	12.80	21.54	10	13.12 ± 1.2	11.48	13.97
Pike-perch (7) (*Sander lucioperca* L.)	10	19.74 ± 0.1	19.48	19.88	NA	NA	NA	NA	10	18.31 ± 2.4	13.94	21.92
Vendace (8) (*Coregonus albula* L.)	10	20.08 ± 0.7	18.9	20.64	11	23.97 ± 1.5	22.21	26.18	10	18.31 ± 1.3	16.42	20.37
Total	74	19.61 ± 1.2	15.88	21.74	66	21.96 ± 2.7	12.80	26.18	72	17.22 ± 2.7	11.48	21.92

^a^ NA—not analyzed, Min.—minimum value, Max.—maximum value, SD—standard deviation.

**Table 2 foods-09-01879-t002:** Fat content in raw, smoked and pickled fish product samples (g/100 g).

Fish Species	Raw				Smoked				Pickled			
	n	Mean ± SD	Min.	Max.	n	Mean ± SD	Min.	Max.	n	Mean ± SD	Min.	Max.
Brown trout (1) (*Salmo trutta m.* *lacustris* L.)	NA ^a^	NA	NA	NA	10	18.81 ± 3.8	13.82	23.98	NA	NA	NA	NA
Common bream (2) (*Abramis brama* L.)	14	2.10 ± 0.6	1.24	3.34	10	3.50 ± 1.8	1.21	5.47	10	4.72 ± 1.1	3.04	6.52
Common perch (3) (*Perca fluviatilis* L.)	10	2.43 ± 1.9	0.89	6.20	10	1.03 ± 0.2	0.72	1.21	10	1.98 ± 2.4	0.01	5.37
Common roach (4) (*Rutilus rutilus* L.)	10	2.82 ± 2.0	1.47	6.86	NA	NA	NA	NA	12	3.11 ± 1.4	1.48	4.74
Common whitefish (5) (*Coregonus lavaretus* L.)	10	3.28 ± 0.3	2.83	3.77	12	3.11 ± 1.5	1.53	5.83	10	3.77 ± 0.9	2.84	5.46
European eel (6) (*Anguilla anguilla* L.)	10	18.20 ± 2.5	14.89	22.33	13	28.16 ± 8.6	18.70	43.12	10	25.04 ± 4.1	18.26	29.72
Pike-perch (7) (*Sander lucioperca* L.)	10	1.11 ± 0.1	1.01	1.26	NA	NA	NA	NA	10	1.72 ± 2.0	0.01	6.02
Vendace (8) (*Coregonus albula* L.)	10	4.46 ± 1.2	2.18	5.33	11	5.34 ± 1.7	2.19	7.38	10	3.04 ± 1.0	1.66	4.45
Total	74	4.76 ± 5.6	0.89	22.33	66	10.54 ± 11.2	0.72	43.12	72	6.11 ± 8.0	0.01	29.72

^a^ NA—not analyzed, Min.—minimum value, Max.—maximum value, SD—standard deviation.

**Table 3 foods-09-01879-t003:** Salt content in raw, smoked and pickled fish product samples (g/100 g).

Fish Species	Raw				Smoked				Pickled			
	n	Mean ± SD	Min.	Max.	n	Mean ± SD	Min.	Max.	n	Mean ± SD	Min.	Max.
Brown trout (1) (*Salmo trutta m. lacustris* L.)	NA ^a^	NA	NA	NA	10	1.09 ± 0.2	0.86	1.36	NA	NA	NA	NA
Common bream (2) (*Abramis brama* L.)	14	0.91 ± 0.1	0.73	1.09	10	1.45 ± 0.2	1.04	1.72	10	2.03 ± 0.1	1.89	2.24
Common perch (3) (*Perca fluviatilis* L.)	10	1.01 ± 0.1	0.89	1.16	10	2.45 ± 0.5	2.07	3.39	10	2.04 ± 0.2	1.82	2.24
Common roach (4) (*Rutilus rutilus* L.)	10	1.11 ± 0.1	0.90	1.24	NA	NA	NA	NA	12	1.76 ± 0.1	1.69	1.88
Common whitefish (5) (*Coregonus lavaretus* L.)	10	0.85 ± 0.1	0.77	0.95	12	1.96 ± 0.3	1.55	2.63	10	1.70 ± 0.1	1.47	1.83
European eel (6) (*Anguilla anguilla* L.)	10	1.34 ± 0.1	1.12	1.48	13	1.94 ± 0.5	0.77	2.46	10	1.69 ± 0.1	1.65	1.76
Pike-perch (7) (*Sander lucioperca* L.)	10	1.01 ± 0.1	0.97	1.09	NA	NA	NA	NA	10	2.02 ± 0.2	1.69	2.29
Vendace (8) (*Coregonus albula* L.)	10	1.02 ± 0.1	0.96	1.13	11	1.61 ± 0.7	1.08	2.93	10	1.92 ± 0.1	1.74	2.06
Total	74	1.03 ± 0.2	0.73	1.48	66	1.76 ± 0.6	0.77	3.39	72	1.88 ± 0.2	1.47	2.29

^a^ NA—not analyzed, Min.—minimum value, Max.—maximum value, SD—standard deviation.

**Table 4 foods-09-01879-t004:** Collagen content in raw, smoked and pickled fish product samples (g/100 g).

Fish Species	Raw				Smoked				Pickled			
	n	Mean ± SD	Min.	Max.	n	Mean ± SD	Min.	Max.	n	Mean ± SD	Min.	Max.
Brown trout (1) (*Salmo trutta m. lacustris* L.)	NA ^a^	NA	NA	NA	10	1.06 ± 0.3	0.64	1.39	NA	NA	NA	NA
Common bream (2) (*Abramis brama* L.)	14	0.51 ± 0.1	0.34	0.75	10	0.28 ± 0.2	0.05	0.54	10	0.96 ± 0.1	0.67	1.11
Common perch (3) (*Perca fluviatilis* L.)	10	0.75 ± 0.5	0.34	1.81	10	0.26 ± 0.2	0.01	0.57	10	1.39 ± 0.8	0.70	2.45
Common roach (4) (*Rutilus rutilus* L.)	10	0.49 ± 0.2	0.20	0.94	NA	NA	NA	NA	12	1.53 ± 0.5	0.99	2.26
Common whitefish (5) (*Coregonus lavaretus* L.)	10	0.39 ± 0.1	0.30	0.47	12	0.07 ± 0.1	0.00	0.22	10	0.36 ± 0.4	0.00	0.88
European eel (6) (*Anguilla anguilla* L.)	10	0.75 ± 0.2	0.41	1.02	13	1.39 ± 0.6	0.45	2.25	10	2.45 ± 0.4	1.58	2.94
Pike-perch (7) (*Sander lucioperca* L.)	10	0.73 ± 0.1	0.65	0.86	NA	NA	NA	NA	10	1.27 ± 0.7	0.40	2.52
Vendace (8) (*Coregonus albula* L.)	10	0.36 ± 0.4	0.09	1.06	11	0.1 ± 0.1	0.00	0.28	10	1.07 ± 0.4	0.57	1.84
Total	74	0.57 ± 0.3	0.09	1.81	66	0.54 ± 0.6	0.00	2.25	72	1.30 ± 0.8	0.01	2.94

^a^ NA—not analyzed, Min.—minimum value, Max.—maximum value, SD—standard deviation.

**Table 5 foods-09-01879-t005:** Water content in raw, smoked and pickled fish product samples (g/100 g).

Fish Species	Raw				Smoked				Pickled			
	n	Mean ± SD	Min.	Max.	n	Mean ± SD	Min.	Max.	n	Mean ± SD	Min.	Max.
Brown trout (1) (*Salmo trutta m. lacustris* L.)	NA	NA ^a^	NA	NA	10	57.13 ± 3.6	52.27	60.67	NA	NA	NA	NA
Common bream (2) (*Abramis brama* L.)	14	77.61 ± 1.7	75.03	80.66	10	72.46 ± 2.1	69.31	74.56	10	71.06 ± 0.6	70.12	72.40
Common perch (3) (*Perca fluviatilis* L.)	10	77.41 ± 2.5	72.82	80.00	10	73.55 ± 1.6	70.91	75.31	10	73.42 ± 2.7	68.55	76.02
Common roach (4) (*Rutilus rutilus* L.)	10	76.93 ± 1.9	73.51	79.51	NA	NA	NA	NA	12	73.83 ± 1.3	71.27	75.39
Common whitefish (5) (*Coregonus lavaretus* L.)	10	75.50 ± 1.1	74.2	77.11	12	72.37 ± 2.6	68.69	76.35	10	74.83 ± 1.1	72.51	76.27
European eel (6) (*Anguilla anguilla* L.)	10	64.36 ± 2.1	61.01	66.67	13	50.54 ± 7.3	38.11	58.84	10	59.37 ± 3.1	53.30	64.55
Pike-perch (7) (*Sander lucioperca* L.)	10	78.43 ± 0.3	78	78.81	NA	NA	NA	NA	10	73.15 ± 1.7	70.48	76.30
Vendace (8) (*Coregonus albula* L.)	10	75.66 ± 1.4	74.72	78.34	11	70.25 ± 2.0	66.94	72.68	10	73.06 ± 1.7	70.20	75.68
Total	74	75.26 ± 4.7	61.01	80.66	66	65.60 ± 10.0	38.11	76.35	72	70.31 ± 9.6	53.30	76.30

^a^ NA—not analyzed, Min.—minimum value, Max.—maximum value, SD—standard deviation.

**Table 6 foods-09-01879-t006:** Energy value in 100 g of raw, smoked and pickled freshwater fish products.

Fish Species	kJ			kcal		
	Raw	Smoked	Pickled	Raw	Smoked	Pickled
Brown trout (*Salmo trutta m. lacustris* L.)	NA	1060.1	NA	NA	253.2	NA
Common bream (*Abramis brama* L.)	409.4	523.8	476.6	97.8	125.1	113.8
Common perch (*Perca fluviatilis* L.)	415.3	420.5	351.3	99.2	100.4	83.9
Common roach (*Rutilus rutilus* L.)	439.9	NA	401.2	105.1	NA	95.8
Common whitefish (*Coregonus lavaretus* L.)	471.1	502.5	469.6	112.5	120.1	112.2
European eel (*Anguilla anguilla* L.)	982.6	1367.9	1163.3	234.7	326.7	277.8
Pike-perch (*Sander lucioperca* L.)	372.4	NA	371.5	89.0	NA	88.7
Vendace (*Coregonus albula* L.)	504.3	602.7	421.2	120.5	143.9	100.6
Total	507.8	764.9	518.6	121.3	182.7	123.9

NA—not analyzed.

**Table 7 foods-09-01879-t007:** Nutrient value provided by 150 g of freshwater fish products.

	(%) RI of kJ			(%) RI of kcal			(%) RI of Protein			(%) RI of Fat			(%) RI of Salt		
	R	S	P	R	S	P	R	S	P	R	S	P	R	S	P
Brown trout (*Salmo trutta m. lacustris* L.)	NA	18.9	NA	NA	19.0	NA	NA	62.9	NA	NA	40.3	NA	NA	27.3	NA
Common bream (*Abramis brama* L.)	7.31	9.35	8.51	7.33	9.38	8.54	59.2	70.2	53.5	4.50	7.50	10.1	22.8	36.3	50.8
Common perch (*Perca fluviatilis* L.)	7.41	7.51	6.27	7.44	7.53	6.29	58.0	68.4	49.6	5.21	2.21	4.24	25.3	61.3	51.0
Common roach (*Rutilus rutilus* L.)	7.85	NA	7.16	7.88	NA	7.19	59.8	NA	50.9	6.04	NA	6.66	27.8	NA	44.0
Common whitefish (*Coregonus lavaretus* L.)	8.41	8.97	8.38	8.44	9.01	8.41	62.3	69.1	58.7	7.03	6.66	8.08	21.3	49.0	42.5
European eel (*Anguilla anguilla* L.)	17.5	24.4	20.8	17.6	24.5	20.8	53.2	55.0	39.4	39.0	60.3	51.5	33.5	48.5	42.3
Pike-perch (*Sander lucioperca* L.)	6.65	NA	6.63	6.67	NA	6.54	59.2	NA	54.9	2.38	NA	3.69	25.3	NA	50.5
Vendace (*Coregonus albula* L.)	9.01	10.8	7.52	9.03	10.8	7.54	60.2	71.9	54.9	9.56	11.4	6.51	25.5	40.3	48.0
Total	9.07	13.6	9.26	9.10	13.7	9.29	58.8	65.9	51.7	10.2	22.6	13.1	25.8	44.0	47.0
RI ^a^	8400 kJ			2000 kcal			50 g			70 g			6 g		

NA—not analyzed; R-raw, S-smoked, P-Pickled; RI- Reference Intake, ^a^ [22].

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
