# Peer review of "Proximal Composition and Nutritive Value of Raw, Smoked and Pickled Freshwater Fish"

_foods, 2020, doi:10.3390/foods9121879_

Round 1
Reviewer 1 Report
Authors have been improved the manuscript following the suggestions of all reviewers.
Reviewer 2 Report
The manuscript has been revised. The authors considered the majority of the comments and suggestions referred by the reviewers. I have no further comments.
This manuscript is a resubmission of an earlier submission. The following is a list of the peer review reports and author responses from that submission.
Round 1
Reviewer 1 Report
The following correction should be addressed before publication.
- Page 1, line 36: add after “…lifestile diseases [2].” the sentence “Nevertheless, toxicologist recommended cautions because fish is an important source of exposure to many contaminants. See “Trace elements in Thunnus thynnus from Mediterranean Sea and benefit–risk assessment for consumers. 2015. Food Additives & Contaminants: Part B, 8, 175–181”, and “Heavy metals content by ICP-OES in Sarda sarda, Sardinella aurita and Lepidopus caudatus from the Strait of Messina (Sicily, Italy). Natural product research, 27(6), 518-523”. These refe
Reviewer 2 Report
Introduction: Author should start with aquaculture importance or nutritional value of those fish. Then, their availability in those lakes to be mentioned. Earlier studies also investigated the body composition of many fish species. So how this study is unique?
Materials and method: Mention the average weight of those fish species.
Line 62: In case Salmo trutta morpha… , why only smoked fish sample was collected? Did they prepared those fish samples or procured from market?
How about the weight of various samples?
Authors only studied few parameters, which are very common. Why authors did not study the presence of heavy metals (if any) in fish samples? At least studying few heavy metals will make the paper interesting.
Why authors used only mid-infra spectrometry? Sample preparation and instrumentation need more description.
The authors could add to the section 2.4 the temperature ramp and the time used to cause chemical andthermal decomposition of the samples, and the wavelength of the Atomic Absorption that have used fordetermination of mercury, howerver this last is the only one possible (253.7 nm).
Reviewer 3 Report
The manuscript is focused on the proximate composition, energy and nutritive value of several fish species with some preservation methods. It is merely a descriptive paper. However, the title and the objetive lead to confusion. It seems that authors developed some NIRS (or MIR) calibrations but they used, I guess, commercial calibrations developed by Foss. In addition, I do not know why MIR spectra were collected and showed but there are in the manuscript nor utility or discussion. At L119-120 it is said that raw spectra are difficult to interpret. That is why often some mathematic treatments are applied to spectra... A high effort is needed to improved the paper, mainly the aim. In my opinion, the title should be something like "Proximal composition and nutritive value of raw, smoked and pickled freschwater fish". The aims can remain as is, and all the MIR statements could be deleted. Then, the manuscript can be resubmitted.
The Figures 1-4 are unnecesary.
L79-95 If the calibrations and ANN are developed by Foss it must be clearly written and the method of AOAC or similar must be referred. Also add the model, manufacturer, city and country of both NIR and MIR
